# HPMCAS-Based Amorphous Solid Dispersions in Clinic: A Review on Manufacturing Techniques (Hot Melt Extrusion and Spray Drying), Marketed Products and Patents

**DOI:** 10.3390/ma16206616

**Published:** 2023-10-10

**Authors:** Leander Corrie, Srinivas Ajjarapu, Srikanth Banda, Madhukiran Parvathaneni, Pradeep Kumar Bolla, Nagavendra Kommineni

**Affiliations:** 1School of Pharmaceutical Sciences, Lovely Professional University, Phagwara 144411, Punjab, India; leander.corrie8@gmail.com; 2Thermo Fisher Scientific Inc., Cincinnati, OH 45237, USA; shankarsrinivasajjarapu@gmail.com; 3Department of Chemistry and Biochemistry, Florida International University, 11200 SW 8th Street, Miami, FL 33199, USA; sband003@fiu.edu; 4Department of Biotechnology, Harrisburg University of Science and Technology, Harrisburg, PA 17101, USA; madhu.parvathaneni@gmail.com; 5Department of Biomedical Engineering, College of Engineering, University of Texas at El Paso, El Paso, TX 79968, USA; 6Center for Biomedical Research, Population Council, New York, NY 10065, USA

**Keywords:** amorphous solid dispersions (ASDs), hydroxypropylmethylcellulose acetate succinate (HPMCAS), bioavailability, manufacturing techniques, patents, clinical trials

## Abstract

Today, therapeutic candidates with low solubility have become increasingly common in pharmaceutical research pipelines. Several techniques such as hot melt extrusion, spray drying, supercritical fluid technology, electrospinning, KinetiSol, etc., have been devised to improve either or both the solubility and dissolution to enhance the bioavailability of these active substances belonging to BCS Class II and IV. The principle involved in all these preparation techniques is similar, where the crystal lattice of the drug is disrupted by either the application of heat or dissolving it in a solvent and the movement of the fine drug particles is arrested with the help of a polymer by either cooling or drying to remove the solvent. The dispersed drug particles in the polymer matrix have higher entropy and enthalpy and, thereby, higher free energy in comparison to the crystalline drug. Povidone, polymethaacrylate derivatives, hydroxypropyl methyl cellulose (HPMC) and hydroxypropyl methylcellulose acetate succinate derivatives are commonly used as polymers in the preparation of ASDs. Specifically, hydroxypropylmethylcellulose acetate succinate (HPMCAS)-based ASDs have become well established in commercially available products and are widely explored to improve the solubility of poorly soluble drugs. This article provides an analysis of two widely used manufacturing techniques for HPMCAS ASDs, namely, hot melt extrusion and spray drying. Additionally, details of HPMCAS-based ASD marketed products and patents have been discussed to emphasize the commercial aspect.

## 1. Introduction

Active pharmaceutical ingredient (API) solubility and bioavailability have been key impediments to the development of more efficient drug delivery techniques for many decades [1]. Various techniques for overcoming this challenge have been offered in the literature, including amorphous solid dispersions (ASDs) [2], nanosuspension [3], polymeric nanoparticles [4,5,6,7,8], liposomes [9], liquid–solid compacts [10,11], solid lipid nanoparticles [12,13,14], self-emulsifying drug delivery systems [15] and other delivery systems [16,17,18,19]. ASDs have been described as the most effective technique for increasing the solubility and dissolution rate, and, as a result, the oral bioavailability of poor water-soluble drugs throughout the last few decades [20]. Because ASDs improve apparent solubility, they also increase apparent permeability, allowing for the oral administration of medicines with low-aqueous solubility [21,22,23,24].

ASDs are single-phase systems in which drug molecules are disseminated or dissolved in one or more polymeric carriers [25]. When compared to pure amorphous drugs, the inclusion of polymeric carriers provides various advantages, including long-term storage stability and improved dissolution capabilities [26,27]. Hydroxypropylmethylcellulose acetate succinate (HPMCAS) has been widely used in ASD formulations as a polymeric carrier. HPMCAS is an effective polymer for the formulation of ASD with dual functionality: solubilization of poorly soluble drugs, promotion of rapid dissolution in the intestinal medium, and prevention of subsequent precipitation of drugs by maintaining supersaturation concentration of the drug molecules [28,29,30,31,32,33]. Several processes, including hot melt extrusion (HME) and spray drying, are used to manufacture ASDs [34]; however, the use of these techniques are dependent on the qualities of the API, desired product attributes and appropriate processing window. Among these approaches, hot melt extrusion (HME) is a supplementary pharmaceutical production technology that is frequently employed by industrial and academic researchers to address API solubility difficulties. In recent years, similar to the lyophilization technique, HME has been used as a drying technique for the solidification or removal of moisture from nanosuspension [35,36]. Though lyophilization is a widely employed drying technique for pharmaceuticals, its usage is associated with time and cost constraints. The main objective of the present review is to discuss HME and spray-drying techniques used to synthesize HPMCAS-based ASDs [37,38,39]. This is followed by providing the marketed formulations of HPMCAS as well as a brief overview of the clinical and patentability aspects.

## 2. Industrial Scale Manufacturing Techniques

### 2.1. Hot Melt Extrusion 

Hot melt extrusion (HME) and spray drying (SD) are the most commonly used techniques for the development of ASDs (Figure 1). Hot melt extrusion is one of the most efficient solid dispersion manufacturing processes. The API and the polymer matrix are blended together in this procedure to create a physical combination that may be extruded under various circumstances [40]. Processing parameters, for example, feed rate, shear force, temperature, die geometry, barrel design, screw speed and other variables should be taken into account when using this method [41]. These parameters could have a considerable impact on the final product’s quality. HME has various advantages over other conventional approaches, including a continuous, one-step, solvent-free operation with fewer processing steps. Furthermore, it does not require compression and can increase the bioavailability as well as increase the biodistribution of the drug at the molecular level [42]. However, some of the restrictions that could limit its application in scaling up and technology transfer are as follows: the need for a larger energy input as compared to other approaches, as well as the likely exclusion of some thermo-labile chemicals due to the high processing temperatures involved [43]. HME has been used successfully to combine various pharmaceutical drug delivery techniques, such as conversion to salts/prodrug approach [44], lipid-based delivery systems [45], immediate and modified release tablets [46] and 3D printing [47,48].

The barrel, feeder and screw parts are the major components of the extruder in HME [49,50]. The extruder barrel is made up of a feeding portion, a venting section and a closed segment design. To soften or reduce the viscosity of the polymer, each part of the barrel can be heated [51]. The feeder facilitates the transmission of material to the barrel. In the HME process, the starvation feeder with a screw speed independent of feed rate is most typically utilized. Screw elements aid in the mixing, transferring, and ultimately pushing of the melt through a die. The screw design makes it easier to set up multiple screw configurations to generate low or high shear. The conveying components aid in pushing the solid material within the barrel, whereas the kneading elements aid in the kneading process [52]. In HME, the crystalline API is dissolved (when the processing temperature is above the melting point of API) or molecularly dispersed (when the processing temperature is below the melting point of API) within the molten polymer owing to the thermal and mechanical energies generated by the rotating screws and heated barrel, respectively [53]. When using HME for ASD, it is necessary to ensure the processing temperature does not degrade the drug or polymer [41]. The HME process can be broken down into the following steps: feeding, melting and plasticizing, conveying and mixing, venting, stripping, and downstream processing [47,48]. While employing HME for ASD production, processing parameters of HME, such as barrel temperature, screw speed, feed rate, barrel design, die geometry and shear force, should be considered [45]. 

### 2.2. Spray Drying

The spray-drying process requires numerous phases involving diverse components. The feed solution/suspension is first introduced into the drying chamber through a nozzle. Droplets are atomized when they escape the nozzle tip and come into contact with the drying fluid, which is hot gas (typically air) inside the drying chamber. The residence duration within the drying chamber is determined by the process parameters and the size of the equipment and can normally last a few milliseconds. At the dynamic droplet surface, energy/mass transfer occurs during the transit through the drying chamber. Finally, using a cyclone, the dried material is separated from the drying area and collected in a collection vessel. The exhaust gases are filtered via HEPA filters. The feed pump utilized is determined by the viscosity of the feed material and the type of atomizer system used [54,55,56,57,58].

When employing this atomization setup, make sure to use a drying chamber with a large enough diameter. Material adherence to the drying chamber walls may limit its application for costly drugs and active moieties. The droplet size distribution produced by the atomization setup dictates the residence time required in the chamber as well as its dimensions. The character of the gas flow (turbulent or laminar) will also influence droplet residence time and final product moisture content. The need for strict inter-batch control of the temperature and humidity of the drying air is especially important for spray drying of amorphous systems [57]. Particles are gathered after drying, utilizing certain design elements and separating mechanisms. The relatively small particle size utilized in medications necessitates this. Particle collection might occur in the bottom of the drying chamber, where it must be scrapped [59]. Bag filters and cyclones are common particle separating devices. In pharmaceutical contexts, typical cyclones are reverse-flow, gas-solid separators in which centrifugal force causes the separation of two phases with differing masses [60].

Spray-drying ASDs are created by evaporating a solvent or a solvent mixture from a drug and polymeric carrier solution. Atomization of the feed stock solution or suspension, droplet-gas contact, droplet drying and particle creation, and separation of dried solid particles from the drying medium (wet drying gas) are all processes in the SD process [61]. To eliminate any leftover solvent, the powder may need to be dried again. While spray drying ASDs, the following critical processing variables should be considered: solution viscosity, total solid content in the spray solution, input and outlet temperatures, nozzle selection and drying gas flow rate [62,63,64,65]. The pros and cons of these techniques are presented in Table 1. 

## 3. Marketed HPMCAS ASD Formulations

The increased Food and Drug Administration (FDA) approval of ASD products in recent years implies that ASD technology can be used to improve the dissolution rate and bioavailability of poorly soluble pharmaceuticals. Over the previous 12 years, the FDA has approved more than 20 ASD products. HPMCAS is used in eight ASD products. The marketed HPMCAS ASDs are presented in Table 2.

### 3.1. Erleada^®^

Erleada^®^ is available in two strengths (60 mg and 240 mg) as an immediate-release tablet formulation of apalutamide [71]. It is indicated for non-metastatic, castration-resistant prostate cancer. Apalutamide is practically insoluble at physiological pH. The preparation process includes, firstly, the preparation of the ASD of apalutamide by using HPMCAS through spray drying [72,73]. The manufacturing process includes pre-blending of the ASD, granulation, extra-granular blending, lubrication, compression and film coating of tablets. A mean C_max_ of 6 µg/mL and 100 µg h/mL was achieved at steady state [74,75].

### 3.2. Trikafta^®^

Trikafta^®^ is an immediate-release triple fixed-dose tablet formulation of elexacaftor, tezacaftor and ivacaftor. It is used in the treatment of cystic fibrosis in patients with F50del allele mutation. All three active ingredients have low solubility [76]. Tezacaftor is a BCS Class II molecule, whereas elexacaftor and ivacaftor can be classified as either BCS Class II or IV molecules based on available information [77]. Elexacaftor has a solubility of less than 0.1 mg/mL in pH buffers ranging from 1.0 to 8.0. Tezacaftor has a solubility of 0.004 mg/mL in pH 1.0, 0.003 mg/mL in pH 4.5 and 0.005 mg/mL in pH 6.8. Trikafta is prepared by a continuous process involving spray drying, blending, granulation, compression of granules into core tablets and film coating of tablets [74]. A new formulation, granules, was later approved in 2021 for the same combination prepared through the same manufacturing process [78,79].

### 3.3. Delstrigo^®^

Delstrigo^®^ is a fixed-dose combination tablet containing Doravirine (100 mg)/lamivudine (300 mg)/tenofovir disoproxil fumarate (300 mg) indicated for the treatment of HIV-1 infection in adult patients with no prior antiretroviral treatment history. Delstrigo^®^ is a bilayer tablet, comprising an ASD formulation of doravirine in the first layer, and lamivudine, tenofovir disoproxil fumarate are separately co-granulated in the second layer [80]. Doravirine is a non-hygroscopic, crystalline powder, which is practically insoluble in water, considered as a Biopharmaceutical Classification System (BCS) class II compound. Lamivudine is soluble in water, sparingly soluble in methanol, slightly soluble in ethanol, and is considered a BCS class III compound. Tenofovir disoproxil fumarate is a slightly hygroscopic powder, slightly soluble in water and sparingly soluble in pH buffers 2.0–8.0. It is considered a BCS class III compound. The manufacturing process consists of spray drying, blending, lubrication, roller compaction, tablet compression and film coating. Briefly, a spray-dried intermediate (SDI) of Doravirine is manufactured by spray drying a solution of HPMCAS and doravirine dissolved in a mixture of water and organic solvent. The SDI is blended and roller compacted after combining with excipients to produce doravirine granules. Lamivudine and tenofovir disoproxil fumarate are combined with excipients, blended and roller compacted to produce the granules. Then, the granules are compressed into a bilayer tablet core and film coated [81].

### 3.4. Symdeko^®^

Symdeko^®^ is a fixed-dose combination of tezacaftor (100 mg)/ivacaftor (150 mg) and ivacaftor (150 mg), indicated for the treatment of patients with cystic fibrosis. Tezacaftor is a crystalline material, practically insoluble in water and buffer solutions from pH 1.0 to pH 9.0. Ivacaftor, active in the combination drug product, has poor dissolution and bioavailability properties in its crystalline form; thus, preparation of an ASD using HPMCAS via spray drying was employed to overcome the solubility and dissolution-limited bioavailability [82]. 

The oral bioavailability of tezacaftor was not determined in humans; however, it was moderate in rats and dogs (~40 to 50%). Absorption of tezacaftor does not vary during fasting or when consumed with fatty foods. In the case of ivacaftor, absorption increases three-fold with fat containing foods. The bioavailability of the crystalline drug and HPMCAS ASD dosed as oral suspension was evaluated using a vehicle composed of methyl cellulose/SLS/water (0.5/0.5/99%). The crystalline form had a bioavailability of 3–6%, whereas solid dispersion showed a bioavailability of around 100%, demonstrating that absorption was dependent on solubility [83].

### 3.5. Orkambi^®^

Orkambi^®^ is a fixed-dose combination immediate-release tablet comprising lumacaftor (200 mg) and ivacaftor (125 mg) for the treatment of cystic fibrosis. Both lumacaftor and ivacaftor are practically insoluble in water with an aqueous solubility of 0.02 mg/mL and <0.05 mg/mL, respectively. Therefore, it is important to develop the amorphous form of active substances in order to improve solubility limited oral bioavailability. In the initial phase-I trials, the bioavailability of a capsule formulation of lumacaftor following oral dose in healthy fasted males showed a higher oral bioavailability with C_max_ and AUC_0-inf_ values 1.4 times higher compared to suspension formulation, and the median T_max_ values for capsule and suspension formulation were 3 h and 4 h, respectively, indicating the capsule formulation was absorbed more slowly than the suspension. Following multiple oral-dose administrations of ivacaftor in combination with lumacaftor, the exposure of ivacaftor increased approximately 2.5 to 3.4-fold when administered with food containing fat [84].

### 3.6. Noxafil^®^ Delayed Release Tablets

The Noxafil^®^ is a delayed release tablet of posaconazole, developed using HME technology to alleviate the inconsistent pharmacokinetics and variable oral bioavailability associated with the oral suspension form of posaconazole. Posaconazole is a broad-spectrum antifungal agent used for prophylaxis and the treatment of fungal infections. A pH sensitive, enteric polymer (HPMCAS) was used to prevent the release of posaconazole in the acidic gastric environment of the stomach, allowing for release in the intestinal site. This delayed release mechanism significantly improved oral bioavailability of posaconazole and achieved higher plasma drug levels with less variability for tablet ASD formulations compared to the suspension formulation. Furthermore, patient’s fed/fasting state effect or consequent administration of medications had no discernible influence on the delayed release of tablet formulation [85,86].

### 3.7. Kalydeco^®^

Kalydeco^®^ is a spray-dried ASD formulation of ivacaftor, indicated for the treatment of cystic fibrosis. When administered in crystalline form, it had an oral bioavailability of 3–6% in rats due to solubility-limited oral absorption. The ASD of ivacaftor was formulated using HPMCAS to overcome the solubility limitations and improve formulation stability. The ASD formulation of ivacaftor exhibited superior solubility (67.4 μg/mL) compared to the solubility of its crystalline polymorph B (1 μg/mL). Furthermore, the ASD formulation demonstrated relative bioavailability of 100% compared to crystalline ivacaftor [87]. After single-dose oral administration to healthy adult volunteers in a fed state, the mean AUC and C_max_ observed were 10,600 ng h/mL and 768 ng/mL, respectively [88,89].

### 3.8. Zelboraf™

Zelboraf™ is the ASD formulation containing vemurafenib in HPMCAS-LF (30:70, *w*/*w*), produced by the solvent/antisolvent precipitation process. The process involves dissolving the drug, ionic polymer (HPMCAS) in N, N-dimethylacetamide, then the solution is precipitated by transferring it into acidified aqueous media. The precipitates are then filtered, washed repeatedly to remove the trace levels of acid and solvent content, vacuum dried and milled to obtain an amorphous powder intermediate known as microprecipitated bulk powder. This amorphous powder provides excellent physical stability with improved oral bioavailability.

The vemurafenib dissolution from its solid dispersions is ~30 times more than crystalline vemurafenib, resulting in approximately five times higher vemurafenib plasma concentrations. During phase-I clinical trials, no tumor regression was observed with conventional vemurafenib formulation at a dose as high as 1600 mg due to the limitations of poor solubility and low oral bioavailability. When patients are treated with ASD of vemurafenib, a substantial tumor regression was achieved as a result of enhanced formulation performance [90].

### 3.9. Incivek^®^

Incivek^®^ is a tablet ASD formulation of telaprevir produced using the spray-drying technique. It is an immediate-release tablet containing 375 mg of telaprevir with a total target weight of 1 g.

Telaprevir is a hepatitis C protease inhibitor that is used to treat genotype 1 chronic hepatitis C in conjunction with peginterferon alfa and ribavirin. Telaprevir is most likely absorbed in the small intestine; nevertheless, absolute bioavailability in humans has not been determined. Telaprevir bioavailability is influenced by food. In phase-3 studies, when a 375 mg tablet formulation was administered to healthy subjects, a three- to four-fold increase in AUC and C_max_ of telaprevir was seen in a fed condition compared to a fasted state. The crystalline form of telaprevir has an aqueous solubility of 4.7 μg/mL and does not ionize between pH 1 and 7, suggesting the poor bioavailability of the drug. Hence, product development focused on converting the crystalline form of telaprevir to a stable amorphous form utilizing HPMCAS and spray-drying technique. The manufacturing process involves the addition of a stabilizing polymer (HPMCAS) to the spray-drying mixture containing drug and organic solvents, then secondary drying of the mixture to remove any residual solvent, tableting with extra granular excipients, compressing the tablets, and finally coating the tablets with a film coating [91]. 

## 4. Mechanism of HPMCAS Drug Release from ASDs

The bioavailability of the drug is enhanced after the drug formulation is present in an amorphous form. A polymer is commonly employed so that the crystalline drug can be converted into an amorphous form diminishing the molecular ability and increasing the glassy temperature [92]. There are various factors that govern the mechanism of drug release from the ASD including the functional charge of the polymer, stereochemistry, selectivity, heterogeneity, etc. [93]. Further, to attain an amorphous nature, the drug crystal lattice is to be broken down, increasing the intermolecular interactions [22]. For a drug to become bioavailable, its dissolution release should be considered [94]. The amount of drug load and formation of colloidal states should be considered. Furthermore, the API should be released from the colloidal state to the molecular state to be allowed to be absorbed from the intestinal medium. There are many mechanisms put forward to understand the release of ASDs in the dissolution medium, such as its conversion into a thick viscous liquid, medium soluble carriers (as in the case of HPMCAS) where the carrier transitions into a colloidal state inhibiting recrystallization to take place [95]. More broadly, it is classified into carrier controlled, dissolution controlled and drug-controlled releases, shown in Figure 2.

After dissolution and passing through the intestinal medium, the transport may be hampered by solubilized API (such as in micelles from endogenous bile salts or surfactants in the formulation) [96]. The role of HPMCAS-facilitating drug release and increasing bioavailability has been studied and elucidated. It was doubtful that smaller particles that resemble intrinsic properties will be ingested. However, ASDs exhibit benefits with regard to bioavailability as contrasted to other supersaturating (solubilizing) drug delivery systems. In an investigation, the authors used an adjusted model and analogous validation in the previously mentioned rat jejunal perfusion model to examine the impact of particles originating from ASDs (progesterone in HPMCAS) [97]. The particles from ASDs assisted the drug mechanism to occur through the intestinal wall. In another study, based on evaluation tests with Caco-2 monolayers and dialysis membranes, Ueda et al. compared the penetration of carbamazepine as ASDs utilizing HPMCAS or Poloxamer 407 as the polymer component. Carbamazepine was more effectively dissolved by Poloxamer 407 than by HPMCAS. Poloxamer 407, on the other hand, reduced drug penetration while HPMCAS boosted it. The authors hypothesized that carbamazepine is self-associated in the HPMCAS solution and that HPMCAS primarily functions as a crystallization inhibitor, limiting molecular mobility. Unlike HPMCAS, Poloxamer 407 generated micelles that encased carbamazepine [98]. Furthermore, various physiological states have an impact on the ASD mechanism, for example, in one study it was found that surface-active compounds, physiological bile salts, are predicted to have an effect on ASD behavior. According to a study by Stewart et al. on itraconazole in HPMCAS, the impact of drug-rich particles on better penetration across the unstirred barrier was lessened as bile-salt concentrations raised [99]. All these studies highlight the fact that HPMCAS in ASDs prevent the crystal forming ability of a drug once it comes in contact with biological membranes. The general mechanism of ASDs is given in Figure 3.

## 5. Clinical Aspects of HPMCAS

The emergence of ASDs is significantly influenced by the physicochemical characteristics of drugs and excipients (polymers). Particle size, Log P, melting temperature (Tm), pKa and glass transition temperature (Tg) are some of the factors that might influence the solubility and stability of formulations that are synthesized especially with HME [100]. The two most important components for every drug’s clinical performance are its solubility and stability, which is interdependent on the polymer being used. In this case, HPMCAS is generally regarded as safe and is the acetate succinate analogue of hydroxy propyl methyl cellulose (HPMC). Numerous clinical studies have shown that HPMC reduces LDL-C without affecting HDL-C or total triglycerides [101]. This indicates the potential of HPMCAS to have the same effects. Overall HPMCAS is known to caused mild adverse effects. Its use as an excipient has been extensively explored, especially in the preclinical setting using beagle dogs. Its use as a filler, solubilizer, ability to change the physiological pH state and alter gut microorganisms has been explored [102]. However, very limited literature is available on the use of HPMCAS and its actual mechanism of action. It would be ideal to know if HPMCAS has any potential effects, such as immunomodulatory and genotoxicity related issues which would only be possible to identify in a clinical setting. This warrants further studies. A list of products using HPMCAS and their clinical outcomes are highlighted in Table 3.

## 6. Patents Related to HPMCAS

There are several patents involved highlighting the role of HPMCAS for the formulation of solid dispersion, granules or tablets. These patents also highlight the versatility of this technique to incorporate poorly soluble drugs belonging to various pharmacological classes. In most of the analysis from the patent search, it was noticed that HPMCAS was used as a coating material. This is because HPMCAS provides sustained release activity and protects drugs from acidic degradation in the stomach. The solvent mixing temperature is an important factor that contributes to the overall amorphous nature and stability of the final formulation, especially after the conversion of crystalline to amorphous form. Several patents have reported a temperature of between −50 and 150 °C. There are also weight equivalent considerations between the hydrophobic segment and hydrophilic segment which contribute to the stability and overall commerciality of the product. Some patents have also highlighted it in this regard. It is noteworthy to mention that HPMCAS has wide applicability and commerciality indicated by the number of patents applied and filed. A list of patents, wherein HPMCAS was used for poorly soluble drugs, is discussed in the subsequent Table 4 in chronological order. 

## 7. Conclusions and Future Perspectives

ASD has been a successful technique for addressing poor water solubility in pharmaceutical compounds; however, the manufacturing processes employed for ASDs are quite challenging, requiring the knowledge of various process-related parameters and product-related activities especially with regard to HPMCAS. While several marketed products are formulated with HPMCAS, it appears that HPMCAS is a current industry preference for solving solubility challenges for oral delivery using ASD using SD and HME techniques that are used for thermolabile drugs and upgrading the drug loading. The increasing number of HPMCAS ASDs on the market demonstrates that HMPCAS is successful for both solubilization as well as stabilizing poorly soluble drugs. Over the past few years, it is proven that HPMCAS-based ASDs are and will continue to be a significant topic of research in the field of formulation development owing to its demonstrated capacity to enhance the oral absorption of poorly soluble drugs. Prediction of drug release from HPMCAS ASDs requires a thorough knowledge of in vitro dissolution testing, deeper understanding of dissolution performance, drug and polymer properties. Utilization of absorption models to capture both gastric and intestinal environments are most useful for HPMCAS-based ASDs. Translation from an in vitro dissolution profile to in vivo oral bioavailability is still a challenge for HPMCAS ASDs. Limited research has been conducted in the literature that dealt with this issue. To anticipate in vivo performance of HPMCAS ASDs, for instance, physiologically based pharmacokinetic (PBPK) modeling or IVIVC (in vitro–in vivo correlation) techniques should be applied. In addition, a deeper mechanistic knowledge of the absorption of APIs in biological systems is beneficial for translational methods. Furthermore, establishing robust and reliable strategies for predicting in vivo exposure from HPMCAS-based ASDs will be an important area in research and development.

## Figures and Tables

**Figure 1 materials-16-06616-f001:**
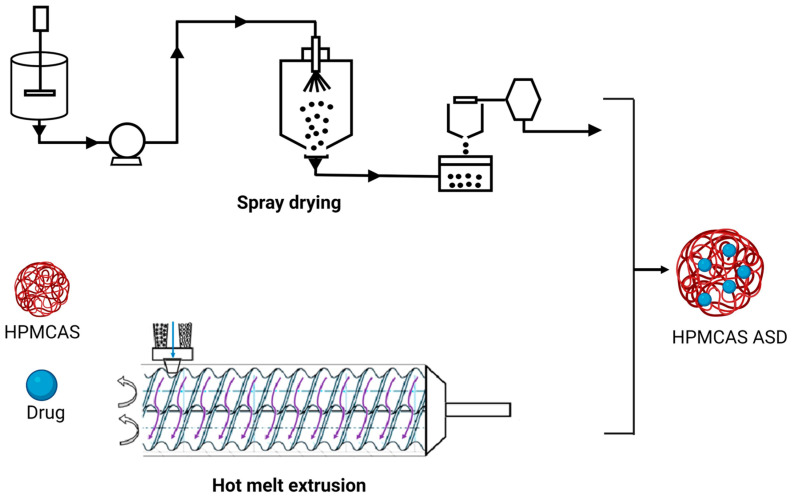
Schematic representation of HME and SD in ASD manufacturing.

**Figure 2 materials-16-06616-f002:**
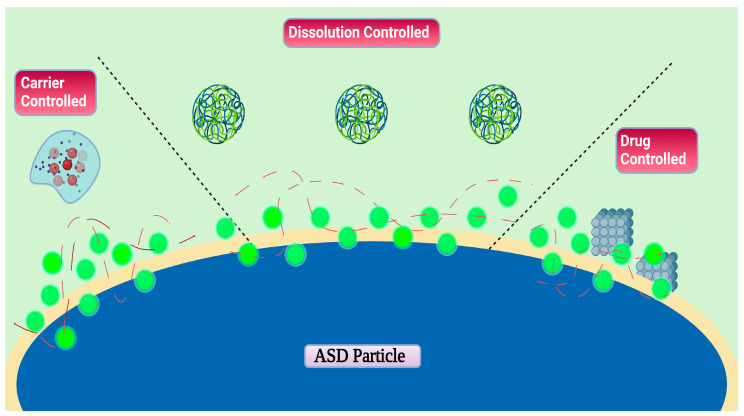
In carrier-controlled cases, drug molecules should permeate through the polymer. Drug-rich particles will develop if dissolved drug levels increase over the amorphous solubility, causing amorphous liquid-phase separation. In dissolution controlled, API and the polymer both dissolve quickly, resulting in the creation of particles that are drug rich. The supersaturated solution might be stabilized by the polymer. In the drug-controlled process, the polymer disintegrates out of the ASD, and the remaining API regulates the rate of disintegration.

**Figure 3 materials-16-06616-f003:**
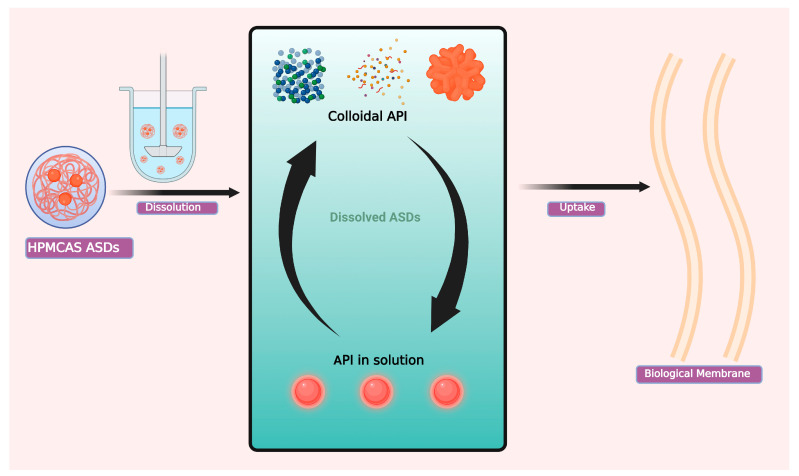
Basic mechanism of increased bioavailability from HPMCAS ASDs.

**Table 1 materials-16-06616-t001:** Advantages and disadvantages of SD and HME in the production of ASDs [66,67,68,69,70].

Process	Pros	Cons
SD	Require a relatively low amount of API, suitable for early-stage screening.	Utilizes organic solvents, which may cause environmental hazards and toxicity, and the solvent recovery process is expensive.
	Non-thermal process suitable for thermo-sensitive APIs.	Require a secondary drying step to remove residual solvent from the ASD particles because the presence of residual moisture may plasticize the material and increase molecular mobility, which may increase the recrystallization propensity of ASDs.
	Provides rapid dissolution due to porous and less dense ASD particles.	Requires additional downstream processing steps such as dry granulation to improve the powder flow.
	Spray dried dispersions can be amorphous even at higher drug loading as opposed to the dispersions of the HME.	Difficult to find solvent or combination of solvent to dissolve both drug and polymer. Often more than one solvent may be required for solubilization.
		Powder handling is difficult because bulk density of spray-dried ASD particle is low.
		As scale changes, particle size of ASD changes, hence, requires some process adaptation before scaling up.
HME	Solvent free technique.	Thermal process, temperature and shear within the barrel may cause thermal degradation.
		May not be suitable for early-stage development due to the requirement of a higher amount of API.
	Ability to produce high-density particles to facilitate downstream processing (tableting).	Additional steps such as blending or granulation may be required before extrusion to achieve uniform feed material consistency.
	The extrudates obtained by HME can be dosed in capsules.	Use of limited number of polymers due to limited availability of thermoplastic or thermally stable polymers.
		Requires addition of plasticizer to allow the physical blend flow through the heated barrel. Over torquing of electric motor or complex viscosity of physical blend greater than 10,000 Pa s necessitate the inclusion of plasticizers.

**Table 2 materials-16-06616-t002:** List of ASD formulations utilizing HPMCAS.

Trade Name	Active (s)	Indication	Manufacturing Technique	Company	Approval Year
Erleada^®^	apalutamide	Treatment of patients with non-metastatic castration-resistant prostate cancer	Spray drying	Janssen Biotech, Inc.	2018
Trikafta^®^	lexacaftor/tezacaftor/ivacaftor and ivacaftor	Treatment of Cystic fibrosis	Spray drying	Vertex Pharms Inc.	2019
Delstrigo^®^	Doravirine/lamivudine/tenofovir disoproxil fumarate	Treatment of HIV-1 infection	Spray drying	Merck	2018
Symdeko^®^	Tezacaftor/ivacaftor and ivacaftor	Cystic fibrosis	Spray drying	Vertex Pharms Inc.	2018
Orkambi^®^	Lumacaftor and ivacaftor	Cystic fibrosis	Spray drying	Vertex Pharms Inc.	2015
Noxafill^®^ Delayed-Release tablet	Posaconazole	Prophylaxis of invasive Aspergillus and Candida infections	HME	Merck Sharp Dohme	2013
Kalydeco^TM^	Ivacaftor	Cystic fibrosis	Spray drying	Vertex Pharms Inc.	2012
Zelboraf^TM^	Vemurafenib	Metastatic melanoma	Solvent/antisolvent precipitation	Hoffmann la roche	2011
Incivek^®^	Telaprevir	Chronic hepatitis C	Spray drying	Vertex Pharms Inc.	2011

**Table 3 materials-16-06616-t003:** Some of the clinical trials relating to using HPMCAS and their outcomes.

Drug	Formulation	HPMCAS conc./Percentageage Used	Technology Used	Human Study	Treatment Duration	Outcome	Reference
Orelabrutinib	Oral tablet	-	SD	OL, NR, MC		Pharmacokinetics data have shown dose linearity	[103]
Vemurafenib	Micro-precipitated bulk powder	60%	SD	RD, OL, CS	84 days	The formulations containing HPMCAS showed comparable and better AUC0-α as compared to the drug	[104]
NTRX-07 (Cannabinoid receptor type 2 (CB2) agonist)	Spray-dried dispersion	-	-	DB, PC	7 days	No treatment related adverse effects Pharmacokinetics of drugs has shown dose linearity	[105]
Posaconazole	Solid oral tablet	-	HME	RD, DB, OL, CS	7 days	HPMCAS reduced the effect of food on Posaconazole which was noticed earlier Greater AUC of the drug	[106]
AFN–1252 (Antimicrobial)	Immediate-release tablet	100 mg	-	SC, DB, RD, PC	7 days	Increase in peak exposure, non-linearly Increase in Cmax	[107]
Ziprasidone	Capsules	20%	SD	RD, OL, CS	NA	Increased extent of ziprasidone absorption Higher Cmax and shorter Tmax Provided improved efficacy	[108]

AUC—Area under curve; C_max_—Concentration maximum; CS—Crossover study; DB—Double blind; HME—Hot melt extrudes; OL—Open label; PC—Placebo controlled; RD—Randomized; NR—Non-randomized; SD—Spray dried; SiD—Single dose; SC—Single center; MC—Multi center; Tmax—Time maximum.

**Table 4 materials-16-06616-t004:** Recent patents on the use of HPMCAS for poorly soluble drugs in chronological order based on the application year.

Patent Number	Drug	Granting Agency	Application Year	Key Claims	Reference
JPH05139964A	Mexiletine Hydrochloride	Japan Patent Office	1991	HPMCAS as coating material Converted into tablets or granules	[109]
US5508276A	Duloxetine	US Patent Office	1994	Enteric layer comprising HPMCAS HPMCAS partially neutralized by ammonium ions	[110]
JPH07330585A	Disopyramide	Japan Patent Office	1994	Tablet prepared from HPMCAS through spray-drying technique	[111]
JP2676319B2	Nicardipine Hydrochloride	Japan Patent Office	1994	Sustained release formulation of Nicardipine	[112]
JPH1160477A	Diclofenac Sodium	Japan Patent Office	1997	Granules of diclofenac sodium prepared along with HPMCAS	[113]
US6171599B1	Efonidipine hydrochloride	US Patent Office	1998	A solid dispersion created using Efonidipine hydrocholride and thermostabilizer such as urea	[114]
US2004228916A1	Poorly soluble drug	US Patent Office	2004	Consist of plasticizer Enteric coating consists of cellulose derivative	[115]
US2008132533A1	Tacrolimus	US Patent Office	2005	Solid dispersion containing HPMCAS and tacrolimus	[116]
US2007275061A1	Metformin	US Patent Office	2006	1:1–20:1 by weight composition of hydrophilic and hydrophobic polymers	[117]
KR101156916B1	Imatinib	Korean Intellectual Property Office	2006	The release retardant comprises HPMCAS Conversion of molten granules into tablets	[118]
KR100881372B1	Cefuroxime axetil	Korean Intellectual Property Office	2007	Granules prepared along with nonionic surfactant along with HPMCAS	[119]
US2010240711A1	NPYY5 Receptor Antagonist	US Patent Office	2008	Solid preparation wherein amorphous stabilizer was composed of HPMCAS	[120]
KR20090086686A	Silymarin	Korean Intellectual Property Office	2008	Improved solubility and dissolution rate Dissolution enhancer is composed of HPMCAS	[121]
KR20130020740A	Duloxetine	Korean Intellectual Property Office	2011	Enteric coating layer composed of HPMCAS Enteric coating layer consists of 2–7% of entire weight	[122]
US2015080399A1	Atazanavir	US Patent Office	2014	Drug retarding agent is HPMCAS and acidifying agent is present in 10–30% *w*/*w*	[123]
JP2015205916A	Phenylalanine Derivative	Japan Patent Office	2015	The orally acting polymer is HPMCAS	[124]
CN106880595A	Empagliflozin	Chinese Patent Office	2015	Solid dispersion containing HPMCAS Consisting of at least two pharmaceutically acceptable excipient	[125]
CN106491612A	Tadalafil	Chinese Patent Office	2015	Solid dispersion consisting of HPMCAS Formulation consisting of a binder, lubricant and disintegrant	[126]
CN106727344A	Apremilast	Chinese Patent Office	2015	Solid dispersion containing HPMCAS Mixed in a solvent temperature of −50 to 150 °C	[127]
CN106539760A	Simeprevir	Chinese Patent Office	2015	Solid dispersion containing HPMCAS Mixing at a ratio of 1:0.1 along with the other polymer	[128]
CN106619521A	Itraconazole	Chinese Patent Office	2016	Solid dispersion containing 55–85% of enteric polymer such as HPMCAS	[129]
CN107281108A	Lesinurad	Chinese Patent Office	2016	Solid dispersion containing HPMCAS	[130]
EP3437637A1	Palbociclib	European Patent Office	2016	Solid dispersion containing HPMCAS Molecular weight of HPMCAS to Palbociclib is 0.1:1	[131]
CN108245487A	Ozanimod	Chinese Patent Office	2016	Solid dispersion containing HPMCAS and ozanimod Mixed at a temperature of −50 to 150 °C	[132]
CN115429885A	NSAIDs for treatment of rheumatism	Chinese Patent Office	2022	Solubility enhancement of NSAIDs	[133]
AU2022263496A1	Bendamustine	Australian Patent Office	2022	Solid dispersion containing HPMCAS	[134]
AU2023202710A1	Apalutamide	Australian Patent Office	2023	Solid dispersion containing HPMCAS	[135]

## Data Availability

Not applicable.

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
