# Peer review of "HPMCAS-Based Amorphous Solid Dispersions in Clinic: A Review on Manufacturing Techniques (Hot Melt Extrusion and Spray Drying), Marketed Products and Patents"

_materials, 2023, doi:10.3390/ma16206616_

Round 1

Reviewer 1 Report

In this manuscript, “HPMCAS Amorphous Solid Dispersions in Clinic: A review on Manufacturing techniques, Marketed Products and Patents” by Corrie et al. reviews several solvent and non-solvent 62 pharmaceutical techniques used to synthesize amorphous SDs. Although, authors had put all literatures together, this work seem premature for publication. Therefore, I would suggest that authors may take at least a major revision before resubmission. Here are the comments and suggestions:

1.     The abstract can be extended.

2.     The literatures in this manuscript seem out of date, authors should collect more recent works.

3.     The conclusions should be revised, and what would be perspectives of this manuscript?

Author Response

In this manuscript, “HPMCAS Amorphous Solid Dispersions in Clinic: A review on Manufacturing techniques, Marketed Products and Patents” by Corrie et al. reviews several solvent and non-solvent pharmaceutical techniques used to synthesize amorphous SDs. Although, authors had put all literatures together, this work seem premature for publication. Therefore, I would suggest that authors may take at least a major revision before resubmission. Here are the comments and suggestions:

Response: Authors thank the reviewer for his valuable suggestions to improve the quality of the manuscript.

  1. The abstract can be extended.

Response: The abstract was revised based on reviewer’s suggestion. The revised abstract is now updated in the revised manuscript.

  1. The literatures in this manuscript seem out of date, authors should collect more recent works.

Response: Authors acknowledge reviewer’s comment. The manuscript has been updated with recently approved formulations, clinical trials, and patent information.

  1. The conclusions should be revised, and what would be perspectives of this manuscript?

Response: Authors thank the reviewer for the suggestion. The conclusion section has been modified and updated with future perspectives.

Reviewer 2 Report

Summary

The manuscript entitled ‘HPMCAS Amorphous Solid Dispersions in Clinic: A review on  Manufacturing techniques, Marketed Products and Patents’ presents an overview of the amorphous solid dispersions (ASDs) containing HPMCAS and some related topics such as preparation methods, approved medicines, related patents.

The authors summarized HPMCAS-based ASDs both from the academic and industrial points of view. The manuscript is well-structured but the novelty of the work is questionable. There are several parts of the manuscript, which are too general. Besides, the title is also misleading because only few sections focus on the HPMCAS-loaded ASDs. For instance, there are a lot of patents referred but not all of them are about ASDs. Furthermore, the properties of HPMCAS are not sufficiently explained from material sciences point of view thus I feel that this manuscript does not fit to the scope of the journal.

For all the above-mentioned reasons, I would recommend to reject this manuscript.

Comments

1. Page 2, line 53: Why only those 4 ASD preparation methods are highlighted? There are several others. I would recommend to highlight only those methods, which are used in the case of the marketed products. Otherwise, you need to mention all the other methods from the literature.

2. Page 2, line 63: ASDs abbreviation could be used instead of amorphous SDs.

3. Graphic representation of the HME and SD methods are very weak. I would recommend to make a better quality from this Figure. It looks like a paint drawing.

4. Section 2 is too general. These things are already summarized in many other publications.

5. Page 5, line 149: ASD could be used instead of ‘solid dispersion’.

6. Page 5, line 151: 20 ASD product? Is it true? Is it exactly 20? I am not sure in that it is correct data.

7. Table 2: ® is missing after the name of Symdeko

8. Section 3.1, 3.2, 3.3…: There are a lot of unnecessary information in these sections from the HPMCAS-based ASD formulation point of view.

9. Page 7, line 241: What is the number 68 in this line? What does it mean?

10. Figure 2: The figure caption is too long. The text in the Figure caption should go to the main text. Some research articles also need to referred relating to this Figure because it is not demonstrated well right now.

11. Figure 3: I feel this Figure is too general but it is depicted too specific with the HPMCAS. There could be other polymers that behave similarly.

12. Page 9, line 324: If melting temperature is defined, maybe the glass transition temperature also should be defined.

13. Table 3: Does it contain all products or just some examples? If the second one it would be better to mention it somewhere.

14. Conclusions: ASD abbreviation should used everywhere. It is not consistent here.

Author Response

Summary

The manuscript entitled ‘HPMCAS Amorphous Solid Dispersions in Clinic: A review on Manufacturing techniques, Marketed Products and Patents’ presents an overview of the amorphous solid dispersions (ASDs) containing HPMCAS and some related topics such as preparation methods, approved medicines, related patents.

The authors summarized HPMCAS-based ASDs both from the academic and industrial points of view. The manuscript is well-structured, but the novelty of the work is questionable. There are several parts of the manuscript, which are too general. Besides, the title is also misleading because only few sections focus on the HPMCAS-loaded ASDs. For instance, there are a lot of patents referred to but not all of them are about ASDs. Furthermore, the properties of HPMCAS are not sufficiently explained from material sciences point of view thus I feel that this manuscript does not fit to the scope of the journal.

For all the above-mentioned reasons, I would recommend to reject this manuscript.

Response: Authors thank the reviewer for the constructive feedback on the manuscript. The title has been revised based on reviewer’s comment as the manuscript exclusively discusses only on two manufacturing techniques (HME and spray drying). The table with patent information was revisited as per reviewer’s suggestion and a patent which is not related to the HPMCAS was removed. The other patent information was retained as it was found to be relevant to the scope of the manuscript. The objective of the manuscript is to detail the application of HPMCAS in the preparation of ASDs through two predominant ASD manufacturing techniques (HME and spray drying) and also to discuss the commercial aspect by discussing the clinical trials, approved drug products and patents. The chemical properties from material science standpoint has been discussed thoroughly in other published papers. Hence, the chemical properties of the polymer were not discussed in this manuscript to not deviate from the objective of the manuscript. However, the mechanism by which HPMCAS works to improve the solubility or drug release was extensively discussed in Section 4 of the manuscript.

Comments

  1. Page 2, line 53: Why only those 4 ASD preparation methods are highlighted? There are several others. I would recommend to highlight only those methods, which are used in the case of the marketed products. Otherwise, you need to mention all the other methods from the literature.

Response: The sentence has been revised based on reviewer’s suggestion.

  1. Page 2, line 63: ASDs abbreviation could be used instead of amorphous SDs.

Response: The sentence has been revised based on reviewer’s suggestion.

  1. Graphic representation of the HME and SD methods are very weak. I would recommend to make a better quality from this Figure. It looks like a paint drawing.

Response: The figure has been updated to a better quality as per the reviewer’s suggestion.

  1. Section 2 is too general. These things are already summarized in many other publications.

Response: The objective of the manuscript is to focus on the commercial aspect of the use of HPMCAS in commercial formulations manufactured through HME and spray drying. Hence, a major discussion on the manufacturing techniques is not discussed which is already available in other published literature.

  1. Page 5, line 149: ASD could be used instead of ‘solid dispersion’.

Response: The sentence has been revised based on reviewer’s suggestion.

  1. Page 5, line 151: 20 ASD product? Is it true? Is it exactly 20? I am not sure in that it is correct data.

Response: Authors thank the reviewer’s comment. Around 25 products have been approved by FDA till date. As the numbers keep changing, the sentence has been revised to more than 20 to avoid any data inaccuracy.

  1. Table 2: ® is missing after the name of Symdeko

Response: Table has been revised accordingly.

  1. Section 3.1, 3.2, 3.3…: There are a lot of unnecessary information in these sections from the HPMCAS-based ASD formulation point of view.

Response: Some of the information has been deleted to focus on the HPMCAS-based ASD.

  1. Page 7, line 241: What is the number 68 in this line? What does it mean?

Response: It was a typographical error due to reference management software. 68 was removed from the sentence.

  1. Figure 2: The figure caption is too long. The text in the Figure caption should go to the main text. Some research articles also need to referred relating to this Figure because it is not demonstrated well right now.

Response: The legend has been revised as per the reviewer’s comment.

  1. Figure 3: I feel this Figure is too general but it is depicted too specific with the HPMCAS. There could be other polymers that behave similarly.

Response: We agree with the reviewer that the figure is general. However, the authors would like to keep the figure in the manuscript as it will provide better understating to naïve readers on this topic.

  1. Page 9, line 324: If melting temperature is defined, maybe the glass transition temperature also should be defined.

Response: Glass transition temperature was defined as per reviewer’s comment.

  1. Table 3: Does it contain all products or just some examples? If the second one it would be better to mention it somewhere.

Response: The title has been revised to include” Some of the Clinical Trials relating to using HPMCAS and their outcomes”.

  1. Conclusions: ASD abbreviation should used everywhere. It is not consistent here.

Response: It has been revised based on the reviewer’s suggestion.

Reviewer 3 Report

1. Title: Change HPMCAS to Hydroxypropylmethylcellulose acetate succinate

2. Abstract: Please add more details about the review section 

3. Introduction: Please add the chemical structure of HPMCAS and explain the difference between HPMCAS and other polymers.

4. Industrial scale manufacturing techniques: Please describe other techniques.

5. Clinical aspects of HPMCAS: Please describe in detail the clinical trials.

6. Patents related to HPMCAS: Please describe in detail the patents.

7. Please add a new section about future prospectives.

Author Response

  1. Title: Change HPMCAS to Hydroxypropylmethylcellulose acetate succinate

Response: Authors thank reviewer for his/her valuable time in reviewing the manuscript. Mentioning the full form would make the title very long. Hence, it is being retained as HPMCAS in the title but abbreviated in the content of the manuscript.

  1. Abstract: Please add more details about the review section 

Response: The abstract was revised based on author’s suggestion. The revised abstract was updated in the revised manuscript.

  1. Introduction: Please add the chemical structure of HPMCAS and explain the difference between HPMCAS and other polymers.

Response: The objective of the manuscript is to focus on the commercial aspect of the use of HPMCAS in commercial formulations manufactured through HME and spray drying. The structures and comparison with other polymers is not discussed as it out of the scope of the manuscript.

  1. Industrial scale manufacturing techniques: Please describe other techniques.

Response: The objective of the manuscript is to focus on the commercial aspect of the use of HPMCAS in commercial formulations manufactured through HME and spray drying. Hence, other manufacturing techniques are not discussed in the manuscript.

  1. Clinical aspects of HPMCAS: Please describe in detail the clinical trials.

Response: Authors acknowledge reviewer’s comment. The information in the clinical trials could be exhaustive which does not add value to the content of the manuscript. Hence, only the relevant data which adds value to the manuscript was included in the table.

  1. Patents related to HPMCAS: Please describe in detail the patents.

Response: Authors acknowledge reviewer’s comment. As the patents include several other aspects which are not related to the scope of the manuscript, only information which is relevant to the manuscript has been included in the tables.

  1. Please add a new section about future prospectives.

Response: Future perspectives has been added to the manuscript as per the reviewer’s suggestion.

Round 2

Reviewer 1 Report

It seems more acceptable now.

Author Response

Reviewer 1:

It seems more acceptable now.

Response: Thank you for accepting the revised version of the manuscript.

Although the authors mentioned that all the formulations had a pH value compatible with the pH of human

Reviewer 2 Report

The manuscript has been developed a lot. It became appropriate for publication. Only few minor changes need to be done before the publication.

1. Abstract: 'HPMCAS' abbreviation should be introduced at its first usage.

2. Introduction: It would be better to rewording the main goal. Now the authors say that 'several solvent and non-solvent pharmaceutical techniques...'. However, it would be better to rather focus on HME and spray drying.

3. It would be nice to complete Table 2. with the applied ASD preparation techniques.

4. Line 346: 'Glass transition temperature' should be written with small letter ('glass transition temperature').

Author Response

Reviewer 2:

The manuscript has been developed a lot. It became appropriate for publication. Only few minor changes need to be done before the publication.

Response: We thank the reviewer for the positive feedback. The minor changes requested have been incorporated in the revised manuscript.

1. Abstract: 'HPMCAS' abbreviation should be introduced at its first usage.

Response: Abbreviation of HPMCAS is provided in the abstract.

2. Introduction: It would be better to rewording the main goal. Now the authors say that 'several solvent and non-solvent pharmaceutical techniques...'. However, it would be better to focus on HME and spray drying.

Response: We thank the reviewer for the comment. The goal of the manuscript has been revised as per the reviewer’s suggestion.

3. It would be nice to complete Table 2. with the applied ASD preparation techniques.

Response: We thank the reviewer for the comment. We have now included the manufacturing technique in Table 2 as per the reviewer’s suggestion.

4. Line 346: 'Glass transition temperature' should be written with small letter ('glass transition temperature').

Response: We thank the reviewer for the comment. The manuscript has been revised as per the reviewer’s suggestion.

Reviewer 3 Report

no comments

Author Response

Reviewer 3:

no comments

Response: Thank you for accepting the revised version of the manuscript.